# A New Set of Golden-Gate-Based Organelle Marker Plasmids for Colocalization Studies in Plants

**DOI:** 10.3390/plants11192620

**Published:** 2022-10-05

**Authors:** Hagen Stellmach, Robert Hose, Antonia Räde, Sylvestre Marillonnet, Bettina Hause

**Affiliations:** Department of Cell and Metabolic Biology, Leibniz Institute of Plant Biochemistry, 06120 Halle (Saale), Germany

**Keywords:** colocalization, Golden Gate cloning, green fluorescent protein, mCherry, plant organelles, protoplasts, *Nicotiana benthamiana* leaves, transient transformation

## Abstract

In vivo localization of proteins using fluorescence-based approaches by fusion of the protein of interest (POI) to a fluorescent protein is a cost- and time-effective tool to gain insights into its physiological function in a plant cell. Determining the proper localization, however, requires the co-expression of defined organelle markers (OM). Several marker sets are available but, so far, the procedure requires successful co-transformation of POI and OM into the same cell and/or several cloning steps. We developed a set of vectors containing markers for basic cell organelles that enables the insertion of the gene of interest (GOI) by a single cloning step using the Golden Gate cloning approach and resulting in POI–GFP fusions. The set includes markers for plasma membrane, tonoplast, nucleus, endoplasmic reticulum, Golgi apparatus, peroxisomes, plastids, and mitochondria, all labelled with mCherry. Most of them were derived from well-established marker sets, but those localized in plasma membrane and tonoplast were improved by using different proteins. The final vectors are usable for localization studies in isolated protoplasts and for transient transformation of leaves of *Nicotiana benthamiana*. Their functionality is demonstrated using two enzymes involved in biosynthesis of jasmonic acid and located in either plastids or peroxisomes.

## 1. Introduction

Due to the large data sets obtained by transcriptomics or proteomic approaches, proteins are frequently identified which have an unknown physiological function. Due to in silico analyses, data, such as the protein sequence, theoretical structure, domain analysis, and assignment to a protein family, are usually available. This is, however, often not sufficient to assign a precise biological function to a protein of interest (POI), resulting in a limitation of the gained knowledge in respect to the research question under study. The physiological function of a protein is directly linked to its localization within the cell. Methods for determining which intracellular compartment a protein is associated with can, therefore, provide clues to its function and contribute significantly to the characterization of a newly identified protein [1,2].

A biochemical method to assign proteins to certain organelles is organelle proteomics. This approach requires the isolation and purification of individual organelle fractions from the cell, which are then analyzed by proteomics methods [3]. Organelle proteomics has contributed to the localization of numerous proteins in the past. The disadvantages of this approach are that (i) proteins that occur in minor amounts in an organelle fraction may not be detected, and (ii) there is still the risk of contamination during purification of the fractions [4,5].

In addition to experimental methods to determine localization, in silico methods through bioinformatics tools may help. These tools are mostly based on sequence analysis and examine the POI for organelle-specific signal sequences [1]. In case of some highly conserved sequences, such as the C-terminal retention sequences of the endoplasmic reticulum (ER) and peroxisomes, the analyses already lead to valuable information. In the case of many N-terminal signal sequences of proteins that are distributed via the secretory transport pathway, such as proteins of the plasma membrane or tonoplast, signal sequences are much more variable and, therefore, often not clearly identifiable, making bioinformatics methods somehow limited [1,6].

The most widely applied experimental strategies used for localization studies are microscopic/cell-biological methods. Among them, immune cytology performed with protein-specific antibodies represents a highly specific and sensitive method to localize proteins within a plant cell. It relies, however, on the generation of specific antibodies, which might be time consuming and costly. Far more often, the GOI is fused to protein-coding regions of genes encoding fluorescent proteins and is transformed into a suitable plant system, such as leaves of *Nicotiana benthamiana* or protoplasts derived from them. This allows a simple and often sufficient approach to identify the subcellular distribution within living cells [7]. Direct visualization of the protein-fluorescent protein fusion may already be sufficient for localization in cell organelles that are morphologically very clearly recognizable, such as the nucleus. In the case of organelles that are morphologically and quantitatively very similar, such as mitochondria, peroxisomes, and Golgi vesicles, direct visualization is not suitable for localization [1,8]. Therefore, a prerequisite for the correct localization is the parallel use of well-characterized proteins whose subcellular localization has already been described. These can be coupled to a different fluorescent protein and are used as markers for specific cell organelles, so-called organelle markers (OM). Subsequent analyses then investigate whether the signals of the OM colocalize with that of the POI [9,10]. In addition to co-transformation, transformation of the gene of interest into established lines stably expressing markers for specific cell organelles is also possible [2,8,11,12]. However, the establishment of such lines might be time-consuming and using these lines requires having plants for all of these lines growing in the greenhouse at the time of the experiment. Here, transient co-expression of fluorescently labeled POIs and OM, using PEG-mediated transformation of isolated protoplasts or *Agrobacterium-tumefaciens*-mediated transformation of leaves combines a favorable cost and time effort with sufficient knowledge gain. Thus, transient expression with OMs represents the current method of choice [1,2].

In plant research, there are several well-established model organisms for which published marker sets are available, such as rice [13,14], maize [15], and *Arabidopsis thaliana* [1,10]. The physiological or developmental conditions in cells of other organisms may differ from those of the plant in which the marker localization was originally determined [15]. Therefore, when using an established marker in multiple organisms, it must be excluded that heterologous expression will lead to localization artifacts or fundamentally disrupt cellular processes [1]. OMs suitable for use in multiple organisms are, therefore, based on highly conserved signal sequences. Moreover, they have to not exhibit enzymatic activities or other functions when introduced into the organism [1]. In some cases, therefore, only parts of the protein sequence, such as signal sequences or membrane anchors, are used in combination with fluorescent proteins [1]. Even nonfunctional proteins altered by mutagenesis that still localize correctly are suitable markers from this perspective. In addition to the selection of marker proteins, aspects of proper marker plasmid construction must also be considered.

Generally, OMs are often based on signal sequences that mediate transport to a specific subcellular compartment. These sequences are conserved to varying degrees and are located at different positions within the amino acid sequence depending on the organelle. If a protein does not have a signal sequence, it will remain in the cytoplasm of the cell after translation [1,16], sometimes enter the karyoplasm via diffusion [17], or is imported into the plastid outer envelope by a transit peptide-independent sorting route [18,19]. The transport of proteins into the nucleus, mitochondria, plastids, and peroxisomes is mediated directly from the cytoplasm via specific signal sequences [20]. Proteins of the endoplasmic reticulum, Golgi apparatus, tonoplast, plasma membrane, and proteins that are secreted undergo transport via the secretory transport pathway of the ER and Golgi apparatus. They have a signal sequence for entry into this transport pathway, as well as other sorting signals [20,21]. When constructing the fusion proteins with the fluorophore, care must be taken to ensure that the fusion does not occur at a terminus where a signal sequence required for correct localization is located [1]. In addition, it has to be ensured that the chosen promoter–terminator combination does not lead to too low and, thus, undetectable, or too high (toxic) expression of the marker.

The application of the to-date established marker sets differs only minimally. Usually, the gene of interest (GOI) combined with the coding region of a fluorescent protein (mainly GFP) is co-transformed together with an OM from the established sets. This procedure requires, however, successful co-transformation of OM and POI constructs into the same cell. In case of low transformation efficiencies, the probability of successful co-transformation might hamper a fast and easy localization study. There are already several “tool boxes” available, which provide vectors harboring both the OM and the site for insertion of a GOI, and are based on restriction enzyme digestion/DNA ligase ligation-dependent cloning [22], multisite gateway-mediated cloning [12], or on improved Gibson assembly with unique nucleotide sequences [23]. Here, we describe the development of new vectors designed with a cloning site for one-step insertion of the GOI by Golden Gate cloning [24,25] (Figure 1). These vectors are usable for colocalization studies with defined OM and circumvent the need for different vectors encoding GOI and marker constructs. As a proof of concept, the vectors were tested for localization of two enzymes of the jasmonic acid (JA) biosynthetic pathway, the ALLENE OXIDE CYCLASE (AOC) located in plastids [26] and the OPDA REDUCTASE 3 (OPR3) located in peroxisomes [27]. With this new set of OM vectors, we provide an easy-to-use and reliable tool for localization studies in plant research.

## 2. Results

In order to obtain markers that correctly label various plant organelles, well-characterized markers were tested after insertion into Golden Gate vectors. Based on the markers developed by Nelson et al. (2007), signal peptides and targeting sequences, as well as full-length protein sequences, were used to direct mCherry into various organelles (Table 1). All sequences were cloned into pICH75099 (level 1 backbone), followed by fusion either N-terminally (plastid and mitochondrial target signals, LTI6b) or C-terminally to mCherry. To adjust the strength of expression, different promoters and terminator combinations were tested. Due to the strength of the 35S promoter used also by Nelson et al. (2007), often a mis-localization was visible due to the strong overexpression, as exemplified for the Golgi marker (Appendix A). Therefore, *NOS* promoter and the *35S* terminator was used for the OM resulting in a weaker expression accompanied by a clear labeling of the organelles addressed (Appendix A). The expression of the inserted GOI occurred under the control of the *NOS* promoter and *NOS* terminator. Before creating the final OM vectors (Appendix A), level 1 constructs were tested in two transient transformation systems: (i) mesophyll protoplasts derived from *N. benthamiana* leaves and transformed by PEG-mediated transformation, and (ii) *N. benthamiana* leaves transformed by infiltration of *A. tumefaciens* harboring the respective construct.

### 2.1. Labeling of Plasma Membrane and Tonoplast

To target mCherry to the plasma membrane, the full-length coding region of the Arabidopsis PIP2A encoding a plasma membrane aquaporin was used [1]. Although this fusion properly labeled the plasma membrane in epidermal leaf cells, the transformation of protoplasts resulted in disintegrated protoplasts, which only poorly survived the standard transformation protocol (Figure 2a). Quantification of living and successfully transformed protoplasts after transformation with this construct showed that significantly less protoplasts survived and were transformed in comparison to transformation of mCherry only (Appendix A). It is assumed that the high level of aquaporin incorporated into the plasma membrane results in an enhanced water uptake, leading to bursting of protoplasts. Of central importance for aquaporin-mediated water transport are two so-called NPA motifs, which are highly conserved in the aquaporin family and are essential for both water transport and specificity [36]. In addition, a lysine at position three of the amino acid sequence (K3) is important, because mutation of this amino acid resulted in significantly restricted water transport [37]. To reduce water transport capability, three mutations of PIP2A were created and their influence on localization and protoplast viability was analyzed. The two NPA sequences were replaced by NPC, and K3 was replaced by alanine (K3A). Not all these mutations, however, improved the survival rate of protoplasts (Appendix A). In conclusion, the aquaporin PIP2A did not appear to be suitable as a marker for use in mesophyll protoplasts.

Therefore, other candidates were tested (Table 1): (i) AHA2 is a plasma membrane resident proton pump involved in the formation of the proton gradient along the plasma membrane [28]; (ii) NPSN12 has already been used within the wave line markers [10]; and (iii) LTI6b, which was established as a plasma membrane marker [29]. The coding sequences of AHA2 and of NPSN12 were fused C-terminally to mCherry, whereas LTI6b, a protein of 54 amino acids, was fused N-terminally to mCherry (mCherry–LTI6b) as the marker was established in this orientation [29]. Out of these three candidates, only LTI6b turned out to be useful as a plasma membrane marker in protoplasts and epidermal cells (Figure 2b), since AHA2 and NPSN12 did not label the plasma membrane in both test systems (Appendix A). Due to the difficulties to use the plasma-membrane-located aquaporin PIP2A as a marker in protoplasts, the formerly established tonoplast marker γ-tonoplast intrinsic protein (γ-TIP)–GFP [1] was excluded from the test, since it functions also as a water channel [38]. Alternatively, other tonoplast-located proteins were selected: (i) the ε-subunit (VHA-E1) of a multi-domain proton pump of the tonoplast [31]; (ii) CHN B, a chitinase specifically found in the vacuole [30]; (iii) the vesicle-associated membrane protein 711 (VAMP7121, [10]; and (iv) CDF1, a tonoplast-specific zinc transporter that mediates zinc transport into the vacuole [30]. For the marker candidates CHN B–mCherry and VAMP711–mCherry, a fluorescence signal corresponding to mCherry could only be detected in the case of a few experiments and could not be assigned to an exact cellular structure (Appendix A). This applied also for VHA-E1–mCherry, which was visible throughout the cell (Figure 2c). Only upon transformation of *CDF1–mCherry*, clear signals were visible in protoplasts and epidermal cells appearing in typical bulges at the vacuolar side of chloroplasts, which were visible by their chlorophyll autofluorescence (Figure 2d). These bulges are indicative for the tonoplast, making CDF1 useful as a tonoplast marker among all the tested candidates.

### 2.2. Labeling of Nucleus, ER, Golgi Apparatus, and Peroxisomes

To visualize the cell nucleus, mCherry was fused to Histone H2A W6 (HTA6) from *A. thaliana* [33,34]. The fusion protein resulted in a clear staining of nuclei in protoplasts and epidermal leaf cells (Figure 3a). Labeling of ER (Figure 3b), Golgi apparatus (Figure 3c), and peroxisomes (Figure 3d) was performed using the markers described by [1] (Table 1) and showed unequivocally the respective structures. It is worth mentioning that, for directing mCherry into the ER, both the first 30 amino acids of WAK2 and the C-terminal retention signal K/HDEL were necessary. The use of KDEL or HDEL did not lead to different localizations, but the omission of the WAK2-target sequence resulted in cytosolic localization of mCherry (Appendix A).

### 2.3. Labeling of Plastids and Mitochondria

The plastid and mitochondrial markers were created by fusing mCherry to an organelle-specific transit peptide (TP). For plastids, the RUBISCO TP (aa 1–79) was N-terminally fused to mCherry [1,39] and resulted in a clear fluorescence signal typical for mCherry (Figure 4a). Superposition with chloroplast-specific chlorophyll autofluorescence showed a clear colocalization of the two detected signals. Based on these results, no further markers for plastids were tested and RUBISCO (1–79)–mCherry was used as a plastid marker for further work. Analogous to the generation of the plastid marker, the mitochondrial markers generated were based on the TP of two proteins localized in mitochondria. Here, either amino acids 1–29 of mitochondrial cytochrome C oxidase IV (COX IV) from *S. cerevisiae* [1,16] or amino acids 1–100 of a subunit of the mitochondrial b-c1 complex, the Rieske protein (mtRi) from *S. tuberosum* [40], were fused N-terminally to mCherry. Fluorescence microscopic analysis of isolated protoplasts and epidermal cells showed a dot-like fluorescence signal expected for fluorescently labeled mitochondria (Figure 4b). However, using aa1–29 from COX IV, the signal in epidermal cells was not always limited to the putative mitochondria but was occasionally visible in membranous structures and in the nucleus (Appendix A). Nevertheless, the strongest signal occurred in the same structures, indicating the fact that both OMs label the same organelles. Therefore, mtRi(1–100)–mCherry was chosen as OM for mitochondria.

All the selected OMs were used to create the final Golden Gate vector set (Appendix A, Table 1). Using two versions of the GFP-encoding sequence (with and without stop codon), the possibility for N- and C-terminal fusions with the POI is given. Moreover, vectors for transformation of protoplasts and leaves are provided, the latter being binary vectors that contain an *Agrobacterium* origin of replication, as well as right and left borders. The gene encoding LacZ has been introduced to allow easy selection of the final clones after insertion of the GOI.

### 2.4. Proof of Concept

To test the usability of the new Golden-Gate-based marker set, we used two enzymes of the JA biosynthetic pathway for localization studies. The AOC catalyzing the formation of the main intermediate of JA biosynthesis, *cis*-12-oxo-phytodienoic acid (OPDA), is known to be localized in plastids [26], whereas the subsequent enzyme, the OPDA reductase 3 (OPR3), is located in peroxisomes [27]. We used the cDNA of AOC2 and OPR3 from *A. thaliana* and cloned them first into the “entry vector”, followed by introduction into four different vectors each. These vectors were used either to show a colocalization with the respective OM or to demonstrate the localization in another organelle upon use of a “wrong” OM, both in protoplasts and in epidermal leaf cells. The AOC2 contains an N-terminal transit peptide; therefore, it was cloned into the vector providing its C-terminal fusion with GFP and the plastid-located OM (pAGH1103 and pAGH1121, see Table 1). AOC2–GFP showed a clear colocalization with the plastid marker in both expression systems (Figure 5a).

Co-expression of AOC2–GFP with mCherry located in mitochondria (mtRi (1–100)–mCherry, pAGH1091 and pAGH1109, see Table 1) revealed, however, a clear separation of both fluorescent signals (Appendix A). Since OPR3 carries a peroxisomal localization signal (SKL), its cDNA was cloned into the vector providing the N-terminal fusion with GFP and the peroxisomal-located mCherry (pAGH1130 and pAGH1136, see Table 1). GFP–OPR3 was visible in small dots in protoplasts, as well as epidermal cells, and colocalized with mCherry specifically targeted to peroxisomes (mCherry–SKL, Figure 5b). Co-expression of GFP–OPR3 with mCherry targeted to the ER (WAK (1–30)–mCherry–KDEL, pAGH1132 and pAGH1138) showed distinct and nonoverlapping signals from GFP and mCherry, both in protoplasts and epidermal cells (Appendix A).

## 3. Discussion

The aim of this work was to generate a new set of Golden-Gate-based OM vectors enabling visualization of a protein of interest with simultaneously selective labeling of various plant organelles, such as plasma membrane, tonoplast, nucleus, ER, peroxisomes, Golgi apparatus, plastids, and mitochondria. The presence of only one Golden Gate cloning site (consisting of two *Bsa*I restriction sites) on the OM plasmid should simplify their use. Two sets of vectors with different plasmid backbones were created to use the vectors for either protoplast transformation by PEG-mediated transfer of plasmid DNA or for transformation of *N. benthamiana* leaves with *A. tumefaciens*. After cloning the gene of interest into the OM vectors, the resulting plasmids will contain both coding sequences—the POI fused to GFP and the OM visualized by mCherry—on one vector, circumventing the need for classical co-transformation.

### 3.1. Selected Plasma Membrane and Tonoplast Proteins Ensure Proper Labeling of These Membranes

The selection of different promoters, terminators, and vector backbones to those used for the well-established OM [1,10] might affect the expression strength and, with this, also the localization of the OM [1,10]. Therefore, the new vectors had to be tested again for their proper localization of the encoded proteins. The potential plasma membrane marker PIP2A–mCherry was used as a full-length protein and fused C-terminally to GFP [1]. Analysis of the marker in leaf epidermal cells revealed the expected picture for a plasma membrane marker. Therefore, PIP2A–mCherry is usable as a plasma membrane marker in leaf epidermal cells of *N. benthamiana*. It was, however, obvious that protoplasts could not be successfully transformed with *PIP2A–mCherry* (see Figure 2a). A negative effect of PIP2A-mediated water transport on protoplast integrity was suspected. Therefore, three mutations were created to minimize water transport: two so-called NPA motifs and the lysine at position three of the amino acid sequence were mutated. NPA motifs are highly conserved in the aquaporin family and are essential for both water transport and specificity but not essential for the expression, intracellular processing, and the basic structure of human aquaporins [41,42]. Mutation of NPA to NPC results in decreased water transport in animal cells [43]. The third mutation was based on mutagenesis data of PIP2A from *A. thaliana*, where exchange of lysine to alanine at position three has been reported to reduce water transport by up to 50% [37]. The introduced mutations do not seem to affect the localization of PIP2A in plant cells, which is different to animal cells, where mutation of NPA sequences sometimes led to mis-localization [43]. Although transformed leaf epidermal cells revealed a very homologous picture of PIP2A wild type and mutants, none of the mutants showed a positive effect on protoplast viability (see Appendix A).

Since PIP2A was not usable as common plasma membrane marker, other candidates, such as LTIP6B, AHA2, and NPSN12, were tested to provide an OM usable for both leaf epidermal cells and protoplasts. Expression of *AHA2–mCherry* and *NPSN12–mCherry* did not result in a consistent labeling of the plasma membrane, although both were working in other applications [10,28]. It is hard to speculate why they did not work in our hands, but at least mCherry–LTIP6B showed an unequivocal label of the plasma membrane. This may be due, in part, to its small size of 54 amino acids [29]. It is tempting to speculate that possibly a higher number of mCherry-labeled protein molecules were integrated into the same area of the plasma membrane than in the case of, for example, AHA2, which has a molecular weight approximately 18 times higher than LTIP6B. Usually, plasma membrane markers are fused C-terminally to the fluorescent proteins, because the signal peptide for entry into the secretory pathway is located N-terminally [44]. LTIP6B was, however, N-terminally fused with mCherry following its previous establishment as OM [29,45]. The N-terminal fusion of the fluorophore seems contradictory at first glance. In cases of post-translational mediated translocation into the ER, however, the position of the signal sequence may vary [44,46] or there may be mechanisms for SRP-independent protein import into the ER [47].

Regarding markers for the tonoplast, four candidate proteins were selected. In contrast to CHN B–mCherry and VAMP711–mCherry, the candidate markers VHA-E1–mCherry and CDF1–mCherry showed a clear fluorescent pattern corresponding to tonoplast labeling; a signal exclusively on the vacuolar membrane visible at the non-peripheral side of the chloroplasts and cytoplasm was clearly recognizable (see Figure 2c,d). CHN B–mCherry and VAMP711–mCherry did not meet this criterion. In isolated protoplasts, however, CDF1–mCherry appeared more suitable than VHA-E1–mCherry, which led to a fluorescence appearing frequently in other subcellular compartments. Since the generated markers should be used for both protoplast and leaf transformation, CDF1–mCherry was subsequently established as a tonoplast marker.

### 3.2. The New OM Properly Labeled Nuclei, ER, Golgi Apparatus, and Peroxisomes

Some established organelle marker sets did not contain a nuclear marker [1] because the nucleus has a morphologically very clearly identifiable shape. In view of the at least slight morphological similarity between ER-visualized nuclei and tonoplast bulbs, the possibility to label the nucleus specifically and reliably seems reasonable. Thus, a nuclear marker was generated using HTA6–mCherry [48]. This marker did not show any mis-localization and reliably visualized the expected morphology of the nucleus. Moreover, the marker did not affect cell viability and it proved to be functional in leaf epidermis cells, as well as in isolated protoplasts (see Figure 3a). Thus, HTA6–mCherry fulfills important criteria of an OM and was subsequently chosen as a nuclear marker.

Labeling of ER, Golgi apparatus, and peroxisomes was performed using the markers described in [1]. Both the Golgi marker MAN1(1–49)–mCherry and the peroxisome marker mCherry–SKL showed the expected punctate fluorescence signal, whereas the ER marker with N-terminal signal peptide (WAK2 (aa1–31)) and C-terminal retention signal H/KDEL revealed a strong label of the ER (see Figure 3b–d). However, in the case of localization of ER-resident proteins, the nature of protein transport into the ER and its retention in the ER cause a problem. Assuming an analogous structure of natural ER-resident proteins harboring an N-terminal signal peptide and a C-terminal retention signal, labeling with GFP either N- or C-terminally could mask the signal essential for correct localization of the POI in the ER. N-terminal sequences for recognition by the SRP occur in all proteins, which are co-translationally transported into the ER. In the case of post-translational protein import into the ER, the position of the signal peptide may vary, and cases of SRP-independent protein import into the ER have been described [44,46]. Considering these issues, an ER marker with a universal cloning site for the GOI is difficult to provide, although the marker set provides the possibility of C- and N-terminal GFP fusion of the POI. The generated ER marker construct is usable to label the ER specifically, but fusion of a GOI, which is assumed to be located in the ER, requires prior knowledge of the position of the signal sequences.

### 3.3. N-Terminal Fusion of TPs to mCherry Led to Proper Labeling of Plastids and Mitochondria

While the ability to detect chlorophyll autofluorescence would be sufficient for the visualization of chloroplasts, other plastid types cannot be visualized based on autofluorescence. Thus, a marker that is suitable for labeling plastids in general seems quite useful [1]. The use of the Rubisco TP enabled successful labeling of plastids in protoplasts and epidermal leaf cells (see Figure 4a). The mitochondrial marker was created using either the TP (aa1–29) of mitochondrial cytochrome C oxidase IV (COX IV) from *S. cerevisiae* [1,16] or the TP (aa1–100) of a subunit of the mitochondrial b-c1 complex, the Rieske protein (mtRi) from *S. tuberosum* [40]. Both TPs were fused N-terminally to mCherry (see Figure 4b). Although both markers labeled dot-like structures (mitochondria) in protoplasts and epidermal leaf cells, use of COX IV(1–29)–mCherry often resulted in an additional fluorescence of the nucleus and along membrane-like structures. This might be due to the combination of promotor and fluorophore used in this study and, therefore, differs from the previously published mitochondrial marker [1].

### 3.4. Proof of Concept for Golden-Gate-Based OM Reveals Their Functionality

The functionality of the dual marker concept was demonstrated using different proteins requiring either C-terminal or N-terminal fluorophore fusions. In addition to the confirmation of the correct localization of OM, some of the created markers were tested as an example for both colocalization as well as for non-colocalization. We used two enzymes involved in the biosynthesis of JA, whereby one enzyme is located in the plastids (AOC [26]) and the other is known to occur in peroxisomes (OPR3 [27]). The cDNAs were amplified and introduced into the Level 1 vectors, followed by its transfer into the final Golden Gate vector (see Figure 1). All the combinations created and tested showed the expected colocalization or non-colocalization and confirmed the ease of use of the newly generated vector set (see Figure 5 and Appendix A).

## 4. Materials and Methods

### 4.1. Cloning of Organelle-Targeted mCherry

Coding sequences of selected organelle-targeted proteins (see Table 1) were cloned using primers listed in Appendix A, except the coding sequence for LTIB6, which was synthesized by Eurofins Genomics (Ebersberg, Germany), and the TP (aa1-29) of mitochondrial cytochrome C oxidase IV (COX IV), which was available as Golden Gate module pAGM3193 [25]. Directed exchange of nucleotides to remove Golden-Gate-relevant sites by targeted mutagenesis or to introduce targeted mutations into marker candidates was performed using the Q5^®^ Site-Directed Mutagenesis Kit (New England Biolabs, Frankfurt am Main, Germany) (primers listed in Appendix A). Level 1 constructs containing the coding sequence for mCherry fused to the respective organelle target sequence or a defined protein were created using the Golden Gate cloning method [25,49]. All final cloning cassettes were assembled into the level 2 vectors pAGM29108 and pAGM29133 to obtain the vectors for protoplast and leaf transformation, respectively. The backbone for protoplast transformation was designed to have a minimal size (2.0 kb) and consists of origin of replication (pMB1 ori) and a bacterial selection marker (kanamycin resistance). In contrast, the backbone for transformation of *N. benthamiana* leaves is a binary vector that contains, in addition to the pMB1 ori and the kanamycin resistance gene, an Agrobacterium origin of replication (pVS1 ori) and right and left borders, resulting in a backbone size of 4.8 kb. Final vectors were transformed into *E. coli* Dh10b and *A. tumefaciens GV3101 pMP90* to obtain plasmid DNA for protoplast transformation or perform leaf transformation, respectively.

### 4.2. Transformation of Protoplasts from N. benthamiana and Determination of Protoplast Vitality

Plasmid DNA was isolated from 45 mL *E. coli* overnight culture using the “NucleoSnap^®^ Plasmid Midi“ Kit (Macherey-Nagel, Dueren, Germany) and purified using precipitation with PEG4000 as described [50]. Protoplast transformation was conducted as published previously [51,52]. Briefly, protoplasts were isolated from the third and fourth leaf of 4-week-old *N. benthamiana* plants by cell wall digestion with 1.5% (*w*/*v*) cellulase Onozuka R-10 and 0.4% (*w*/*v*) macerozyme R-10 (both from Yakult, Tokyo, Japan) in 0.4 M mannitol, 20 mM KCl, 20 mM MES (pH 5.7), 110 mM CaCl_2_, and 0.1% (*w*/*v*) BSA. After purification, 200 µL of a solution containing 10^5^ protoplasts per mL were incubated with 10 µg plasmid and 40% (*w*/*v*) PEG in 0.2 M mannitol and 100 mM CaCl_2_ for 5 min. After sedimentation at 200× *g* for 1 min, cells were resuspended in 200 µL buffer (0.5 mM mannitol, 20 mM KCl, 4 mM MES, pH 5.7) and incubated at room temperature overnight.

To determine the vitality of protoplasts after transformation, protoplasts were stained with fluorescein diacetate (Sigma-Aldrich, Darmstadt, Germany) according to [53] and analyzed using an epifluorescence microscope AxioImager (Zeiss GmbH, Oberkochen, Germany) equipped with the proper filter combination. Rate of vital cells was determined from at least 300 protoplasts per biological replicate by counting living cells showing strong green fluorescence upon excitation with 490 nm and dead cells, which were nonfluorescent.

### 4.3. Transformation of Leaves from N. benthamiana

Constructs were transformed into *Agrobacterium tumefaciens* strain *GV3101 pMP90* using electroporation. Cells obtained from an overnight culture were resuspended in 10 mM MgCl_2_, 10 mM MES, pH 5.7 supplied with 200 µM acetosyringon to an OD of 0.5 and directly infiltrated into leaves of 4-week-old *N. benthamiana* plants [54]. Infiltrated areas were analyzed after 48 h.

### 4.4. Microscopy

Infiltrated leaf disk and transformed protoplasts were analyzed by confocal laser scanning microscopy using an LSM780 or LSM880 (Zeiss GmbH). Upon excitation of 488 nm, GFP fluorescence and chlorophyll autofluorescence were recorded at 493–531 nm and 661–735 nm, respectively. The fluorescence of mCherry was recorded at 580–642 nm using excitation of 561 nm. To visualize the cell shape of epidermal cells after transformation of leaves, a bright-field image was recorded using light of 488 nm.

## 5. Conclusions

The marker set established in this work includes markers for the plasma membrane, tonoplast, nucleus, ER, Golgi apparatus, peroxisomes, plastids, and mitochondria. The provided vectors enable colocalization studies in protoplasts and leaf cells of *N. benthamiana*. Based on well-established OM, all OM vectors were tested and showed, in some cases, mis-localizations or affected strongly the viability of protoplasts. However, for all organelles, reliable markers were selected. The generated marker plasmids allow easy and fast localization experiments due to the two simple cloning steps on the one hand, and having the fusion of the POI with the fluorophore and the marker on one vector, omitting the need for co-transformation of two plasmids, on the other hand. The creation of such a toolbox for the Golden Gate cloning method is, therefore, a useful addition to the existing vectors, which are based on different cloning systems, such as Gateway cloning and Gibson assembly.

## Figures and Tables

**Figure 1 plants-11-02620-f001:**
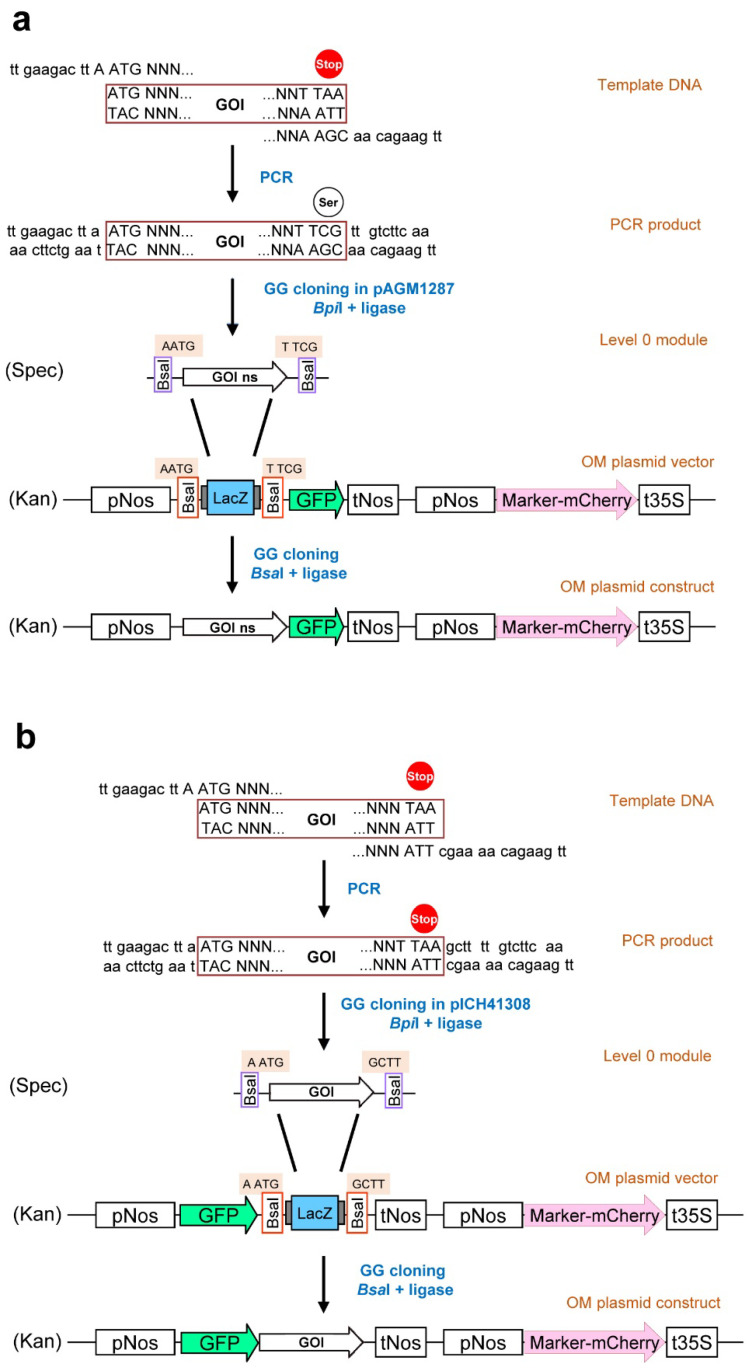
Cloning strategy to obtain protein of interest (POI)–GFP fusions in vectors containing organelle markers (OM). (**a**) C-terminal fusion of POI with GFP. (**b**) N-terminal fusion of POI with GFP. The coding sequence of the POI (GOI) obtained by PCR and extended by the *Bpi*I-recognition sites has to be cloned into level 1 vector pAGM1287 or pICH41308 depending on the envisaged N- or C-terminal fusion with GFP, respectively. As second step, *Bsa*I has to be used to transfer the gene of interest into the final level 2 vector containing the coding sequence of GFP and the selected OM. ns = no stop.

**Figure 2 plants-11-02620-f002:**
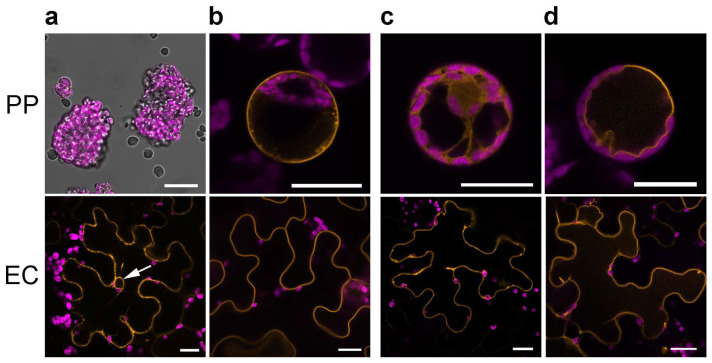
Subcellular localization of proteins located in plasma membrane (**a**,**b**) and tonoplast (**c**,**d**). (**a**) PIP2A–mCherry; note the collapse of mesophyll protoplasts after transformation with the construct encoding PIP2A, whereas it labels also the endomembrane system in epidermal leaf cells (arrow). To visualize the collapsed protoplasts, the bright-field image was added. (**b**) mCherry–LTIP6B; the plasma membrane is clearly labeled in both protoplast and epidermal cells. (**c**) VHA-E1–mCherry; although tonoplast appears to be labelled in epidermal cells, the orange fluorescence appears diffuse within mesophyll protoplasts. (**d**) CDF1–mCherry; transformation with *CDF1::mCherry* resulted in a strong label at the vacuole membrane. Fluorescence of the protein fusion with mCherry is shown in orange, whereas the autofluorescence of chloroplasts is depicted in magenta. PP—protoplast, EC—epidermal cell. Bars represent 20 µm in all micrographs.

**Figure 3 plants-11-02620-f003:**
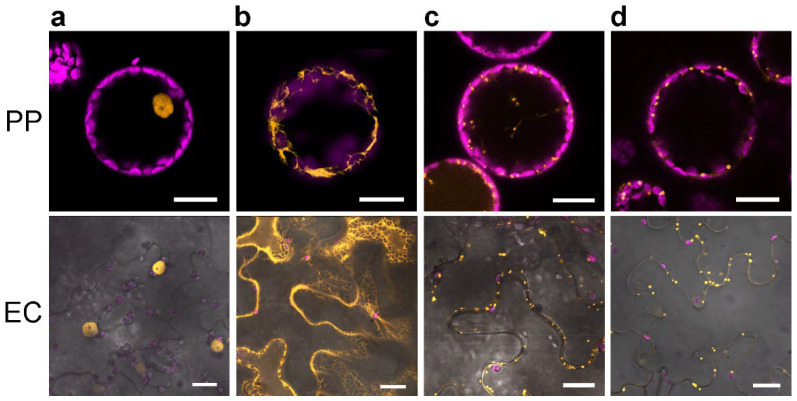
Subcellular localization of proteins located within the nucleus (**a**), endoplasmic reticulum (**b**), Golgi apparatus (**c**), and peroxisomes (**d**). (**a**) HTA6–mCherry. (**b**) WAK2(1-30)–mCherry-KDEL. (**c**) MAN1(1-49)–mCherry. (**d**) mCherry–SKL. Fluorescence of the protein/signal peptide fusion with mCherry is shown in orange, whereas the autofluorescence of chloroplasts is depicted in magenta. To better visualize the location of the respective organelles in epidermal cells, the bright field image was added. PP—protoplast, EC—epidermal cell. Bars represent 20 µm in all micrographs.

**Figure 4 plants-11-02620-f004:**
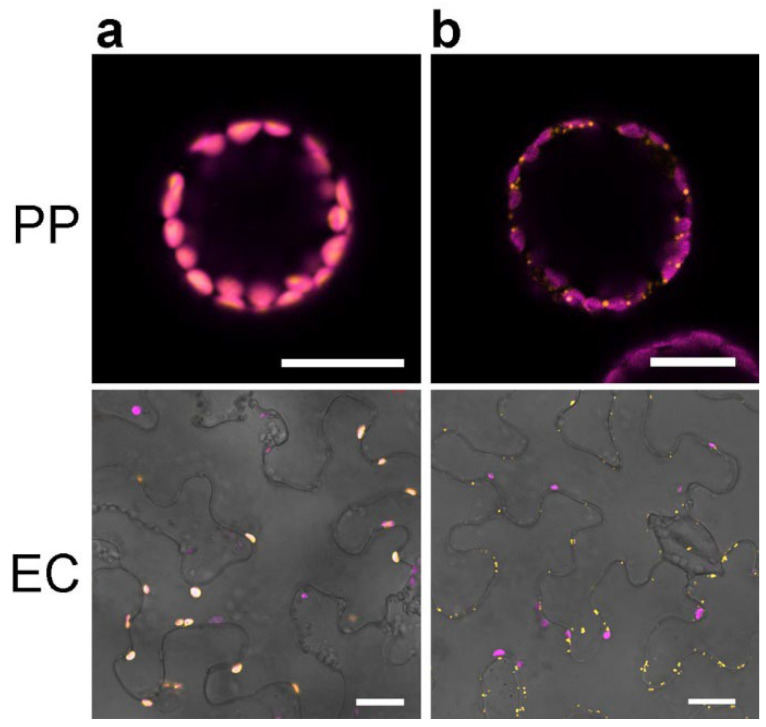
Subcellular localization of proteins located within plastids (**a**) and mitochondria (**b**). (**a**) Rubisco(1–79)–mCherry. (**b**) mtRi(1–100)–mCherry. Fluorescence of the transit peptide fusion with mCherry is shown in orange, whereas the autofluorescence of chloroplasts is depicted in magenta. To better visualize the location of the respective organelles in epidermal cells, the bright-field image was added. PP—protoplast, EC—epidermal cell. Bars represent 20 µm in all micrographs.

**Figure 5 plants-11-02620-f005:**
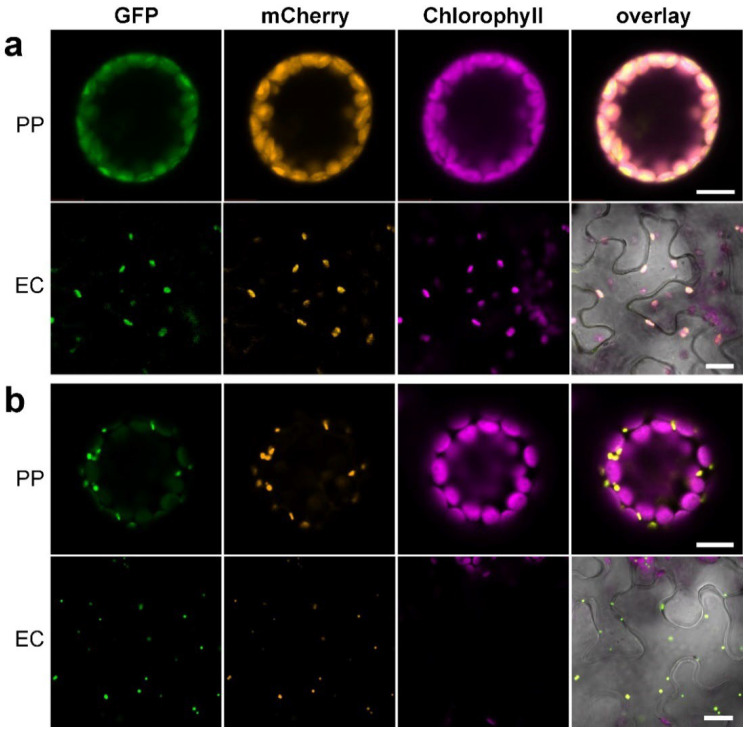
Colocalization of AOC2 (**a**) and OPR3 (**b**) with the respective markers. (**a**) The coding sequence of *AOC2* was cloned into the final OM (pAGH1103 and pAGH1121), which led to C-terminal fusion of AOC2 with GFP and contained the marker for plastids (Rubisco (1–79)–mCherry). (**b**) The coding sequence of *OPR3* was cloned into the final OM (pAGH1130 and pAGH1136), which led to N-terminal fusion of OPR3 with GFP and contained the marker for peroxisomes (mCherry–SKL). Fluorescence of the target protein fused to GFP is shown in green, whereas the OM (mCherry) is shown in orange. The autofluorescence of chloroplasts is depicted in magenta. The overlay shows all three channels; for epidermal cells, the bright-field image was included. Note the overlay of GFP and mCherry indicates colocalization of the target protein with the OM. PP—protoplast, EC—epidermal cell. Bars represent 10 µm for protoplasts and 20 µm for epidermal cells.

**Table 1 plants-11-02620-t001:** Overview about the proteins/genes selected as putative organelle markers.

Candidate		Organism	Ref.	Construct No. (Protoplasts)	Construct No. (Leaves)
**Plasma membrane**					
*Plasma membrane* *intrinsic protein 2A*	PIP2A	*A. thaliana*	[1]		
*H(+)-ATPase 2*	AHA2	*A. thaliana*	[28]		
*Novel plant snare 12*	NPSN12	*A. thaliana*	[10]		
*Low temperature induced* *protein 6b*	LTI6b	*A. thaliana*	[29]	**pAGH1107** (C) ^1^	**pAGH1125** (C)
**Tonoplast**					
*Chitinase B*	CHN B	*N. benthamiana*	[29,30]		
*V**-ATPase* ε-subunit (E1)	VHA-E1	*A. thaliana*	[31]		
*Vesicle associated membrane protein 711*	VAMP711	*A. thaliana*	[10]		
*Cation diffusion facilitator 1*	CDF1	*A. thaliana*	[32]	**pAGH1108** (C)	**pAGH1126** (C)
**Nucleus**					
*Histone H2A W6*	HTA6	*A. thaliana*	[33,34]	**pAGH1104** (C)	**pAGH1122** (C)
**Endoplasmatic reticulum (ER)**					
*Wall-associated kinase2* (aa 1–31) *+ mCherry-KDEL*	WAK2(1–30)-mCherry-KDEL	*A. thaliana*	[1,35]	**pAGH1106** (C) **pAGH1132** (N) ^1^	**pAGH1124** (C) **pAGH1138** (N)
*Wall-associated kinase2* (aa 1–31) *+ mCherry-HDEL*	WAK2(1–30)-mCherry-HDEL	*A. thaliana*	[1,35]	**pAGH1105** (C) **pAGH1131** (N)	**pAGH1123** (C) **pAGH1137** (N)
**Golgi**					
*α-1,2-Mannosidase* (aa 1–49)	*Gm*Man1(1–49)	*Glycine max*	[1]	**pAGH1102** (C)	**pAGH1120** (C)
**Peroxisome**					
*mCherry-SKL* (PTS1)	mCherry-SKL		[1]	**pAGH1130** (N)	**pAGH1136** (N)
**Plastid**					
*Ribulose-1,5-bisphosphate-carboxylase/-oxygenase* (aa 1–79)	Rubisco(1–79)	*N. benthamiana*	[1]	**pAGH1103** (C)	**pAGH1121** (C)
**Mitochondrium**					
*Mitochondrial Rieske protein* (aa 1-100)	Ri(1–100)	*S. tuberosum*	[33]	**pAGH1091** (C)	**pAGH1109** (C)
*Cytochrome c oxidase IV* (aa 1–29)	COXIV	*S. cerevisiae*	[1]		

^1^ (C) and (N) indicate whether the vector enables C-terminal or N-terminal fusion of the POI with GFP.

## Data Availability

The vectors described in this study will be deposited to Addgene.

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
