# Peer review of "A New Set of Golden-Gate-Based Organelle Marker Plasmids for Colocalization Studies in Plants"

_plants, 2022, doi:10.3390/plants11192620_

Round 1

Reviewer 1 Report

The work generated and validated a new set of Golden Gate-based OM vectors to allow for visualization of a protein of interest fused with GFP and the OM visualized by mCherry simultaneously, in both Nicotiana benthamiana leaves and Arabidopsis protoplasts. The newly designed vectors are easy to use with a cloning site for one-step-insertion of the GOI by Golden Gate cloning, which circumvented classical co-transformation, and will facilitate functional studies of POI particularly for localization studies.

One question is why two reported plasma membrane markers, AHA2 and NPSN12, fail to work in this study?

Other issues:

Table 1 should be in three-line format

Fig S5: delete “PP – protoplast, EC – epidermal cell” in the legend

Fig S7 “A” and “B” should not be capitalized.

Author Response

Thank you very much for your efforts to evaluate our manuscript and giving us advise to improve it. We carefully inspected all concerns and have prepared a revision by addressing all concerns and comments as outlined below:

One question is why two reported plasma membrane markers, AHA2 and NPSN12, fail to work in this study?

Answer: The question, why two well-known plasma membrane markers, AHA2 and NPSN12, failed to work in our study is hard to answer. Since we tested several candidates and one of them (LTIP6) worked very well, we did not analyze, why the other two did not work. Listing some possible reasons would just be speculation. Nevertheless, we included or changed the following sentences within the discussion (page 10, lines 353-356): “Expression of AHA2-mCherry and NPSN12-mCherry  did not result in an consistent labeling of the plasma membrane, although both were working in other applications [10,28]. It is hard to speculate, why they did not work in our hands, but at least mCherry-LTIP6B showed an unequivocal label of the plasma membrane.”

Other issues:

Table 1 has been re-formatted.

Legend to Fig. S5 has been corrected.

Figure S7: The “A” and “B” were replaced by “y” and “b”.

Reviewer 2 Report

In this work authors present a new set of organelle marker plasmids for plants. The manuscript is well written and results are clear. It will be great to use these plasmids in co-localization studies when they become available for order in the Addgene repository. I recommend to accept this manuscript after minor revision of small typos and errors in text. For example, at line 517 "Figure S6" should be corrected to "Figure S7".

Author Response

We thank this reviewer for the valuable comments on our manuscript. We checked the manuscript for typos and errors and corrected it, e.g. at line 517 (now line 523) as advised.